# Beyond Negativity: Re-Analysis and Follow-Up Experiments on Hope Speech Detection

**Neemesh Yadav, Mohammad Aflah Khan, Diksha Sethi & Raghav Sahni**
Indraprastha Institute of Information Technology
New Delhi, India
`{neemesh20529,aflah20082,diksha20056,raghav20533}@iiitd.ac.in`

## Abstract

Health experts assert that hope plays a crucial role in enhancing individuals' physical and mental well-being, facilitating their recovery, and promoting restoration. Hope speech refers to comments, posts and other social media messages that offer support, reassurance, suggestions, inspiration, and insight. The detection of hope speech involves the analysis of such textual content, with the aim of identifying messages that invoke positive emotions in people. Our study aims to find computationally efficient yet comparable/superior methods for hope speech detection. We also make our codebase public here.

## 1 Introduction

With the rise of the internet, there has been a significant increase in the number of people seeking support online. Social media comments and posts have been analyzed using tools like hate speech recognition and abusive language detection to limit the spread of negativity. However, these studies primarily focus on analyzing negativity, neglecting the importance of promoting encouraging and supportive online content as positive reinforcement. Our project addresses this gap by identifying hope speech that promotes positivity online. We propose building classifiers using different embeddings & traditional machine learning and deep learning models, whose performance is comparable to the top submissions in two shared tasks Palakodety et al. (2019), Chakravarthi (2020) which often use fairly convoluted architectures. Our study shows that simpler traditional machine learning models, paired with appropriate input representations, can perform as well as complex deep learning models, and suggests considering this approach to save time and resources in model development. Our project has several downstream applications, such as utilizing the classifier to curate more data that can be used to train generative models. These models can then be deployed in toxic online settings to spread positivity and create a more supportive environment.

## 2 Related Work

This study examines the concept of hope speech and the various models used for detecting it. Prior research, such as that by Palakodety et al. (2019), focused on analyzing trends in YouTube comments during times of political tension between India and Pakistan. However, this research defined hope speech as "web content which plays a positive role in diffusing hostility on social media triggered by heightened political tensions," which is limited in scope. Our study uses the broader and more inclusive definition of hope speech provided by Chakravarthi (2020), which defines it as "YouTube comments/posts that offer support, reassurance, suggestions, inspiration, and insight." We utilize the English subset of the HopeEDI dataset provided in the same work.

The inaugural Language Technology for Equality, Diversity, and Inclusion Workshop (LT-EDI-2021) encompassed a shared task aimed at the detection of hope speech, as documented in a subsequent paper Chakravarthi & Muralidaran (2021). The paper detailed the models utilized and their effectiveness, with notable success observed in deep learning-based architectures, such as BERT Devlin et al. (2019) and XLM-RoBERTa Conneau et al. (2019), employed in conjunction with features like an Inception module Szegedy et al. (2015) over two different sources of embeddings (Huang & Bai (2021). In contrast, traditional machine learning approaches were found to be comparatively

| Model[1] | Weighted F1 |
|---|---|
| Shared Task Rank 1 | 0.93 |
| MLP (better-pca) | 0.9262 |
| Logistic Regression (better-no-pca) | 0.9217 |
| Perceptron (better-no-pca) | 0.9204 |
| XGBoost (better-no-pca) | 0.9183 |

Table 1: Three-way classification results

| Model | Macro F1 |
|---|---|
| Shared Task Rank 1 | 0.550 |
| HateBERT | 0.7597 |
| BERT | 0.7552 |
| BERTweet (For Augmented Data) | 0.7597 |
| BERT (For Augmented Data) | 0.705 |
| HateBERT (For Augmented Data) | 0.6872 |

Table 2: Two-way classification results

inferior and fine-tuning was opted for by Hossain et al. (2021) due to this lag. However, our findings indicate that the utilization of appropriate embeddings can obviate the need for fine-tuning. The subsequent iteration of the Workshop (LT-EDI-2022) once again had a shared task focused on hope speech detection, but with the inclusion of datasets for Spanish and Kannada languages. Notably, the evaluation metric was updated from Weighted F1 to Macro F1, as the former could potentially yield inflated scores due to class imbalance Chakravarthi et al. (2022).

## 3 METHODOLOGY

The experimental dataset used in this study was obtained from Chakravarthi et al. (2022). The research employed various word embedding techniques, such as GloVe Pennington et al. (2014), FastText Mikolov et al. (2018), word2vec Mikolov et al. (2013), TF-IDF, and Sentence-BERT Reimers & Gurevych (2019), to capture both non-contextual and contextual representations of the data. Several machine learning models were initially applied using the different embeddings on the three-way classification task (Hope Speech v/s Non Hope Speech v/s Non English). The top-five best-performing models were selected, and additional deep-learning models were tested on the two-way classification task (Hope Speech v/s Non Hope Speech). Details about all the models and their parameters are listed in A.3. In addition, data augmentation techniques, including Random Word Insertion, Swapping, and Deletion, were implemented to address the imbalanced nature of the original dataset. Details regarding the original and augmented dataset statistics are presented in A.1.

## 4 RESULTS

Tables 1 and 2 report the Weighted F1 scores for the three-way classification task and Macro F1 scores for the two-way classification task. The change in metric between tasks follows the original paper. Our results for the first shared task just lag by less than 0.8% in terms of Weighted F1 while being substantially faster and computationally inexpensive to train. In the second shared task, we see that simple finetuning beats the top-ranked models by around 20% in terms of Macro F1. Additionally, we found data augmentation to impact model performance positively.

## 5 CONCLUSION

Our study suggests that simpler traditional machine learning models can offer comparable or even better performance when paired with appropriate input representations than complex deep learning models. Training deep learning models typically requires fine-tuning a large number of parameters, which can be a time-consuming and resource-intensive process. In contrast, we show that embeddings from off-the-shelf pre-trained models combined with traditional machine learning models achieve similar performance while allowing for faster prototyping. Hence it is worthwhile to consider exploring simpler and lighter models as an initial step in the modeling process rather than immediately resorting to deep-learning models. This approach can save time and computational resources without sacrificing model accuracy. Our models offer significantly faster inference, allowing real-time hope speech detection on social media platforms. Similarly, the efficacy of this approach can be used for other such related tasks, such as sentiment classification and hate speech detection, which can be explored in future works.

ACKNOWLEDGEMENTS

We express our gratitude to Sarah Masud for her valuable feedback on the initial drafts of our paper. We also extend our thanks to Dr. Jainendra Shukla for providing guidance during the early stages of this work when it was just a course project. Additionally, we acknowledge the invaluable suggestions from the reviewers, which have greatly contributed to the improvement and refinement of our paper.

URM STATEMENT

The authors acknowledge that all the authors of this work meet the URM criteria of ICLR 2023 Tiny Papers Track.

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

# A  APPENDIX

## A.1  DATASET DETAILS

| Class | Sample |
|---|---|
| Hope Speech | all lives matter without that we never have peace so to me forever all lives matter |
| | im so proud for her |
| Non Hope Speech | its not that all lives dont matter |
| | that would be uhhh pure mean yes |
| Non English | casa la femmenwest villagen 2008nnyc |
| | ¡user¿ tran aah its never too late uh can if u really want tonnyeh mat socho zindagi me kya hoga kuch nhi hoga to tajurba hoga ie do not think what will happen in life |

Table 3: Dataset Samples.

| Class | Number of Samples |
|---|---|
| Hope Speech | 2,484 |
| Non Hope Speech | 25,940 |
| Non English | 27 |

Table 4: Original dataset statistics.

| Split | Hope Speech | Non Hope Speech | Non English |
|---|---|---|---|
| Train | 1962 | 20778 | 22 |
| Dev | 272 | 2569 | 2 |
| Test | 250 | 2593 | 3 |

Table 5: Dataset statistics for each split (train, dev and test).

In the two-way classification task the statistics for Hope Speech and Non Hope Speech Class remain the same while the Non English class is dropped similar to how the Shared Task was updated in the second iteration.

| Class | Number of Samples before Augmentation | Number of Samples after Augmentation |
|---|---|---|
| Hope Speech | 2,484 | 21,582 |
| Non Hope Speech | 25,940 | 25,940 |

Table 6: Dataset statistics before and after data augmentation.

The augmentation was only done for the two-way classification task (Shared Task 2); hence, the Non-English class is not augmented.

## A.2 Embeddings used for the experiments

These embedding prefix codes, along with their names. A pca/non-pca suffix signifies whether PCA was used to reduce dimensions. We request the reader to refer to Table 7 for this subsection.
We chose these 6 different embeddings as they cover the most popularly used non-contextual and contextual embeddings. We anticipated SentenceBERT Embeddings to be significantly better than the non-contextual embeddings, and we observed the same in our results. We agree that there are several other embeddings that we could've tried, but the results with these embeddings alone were quite close to the SOTA models, and hence we chose to stick with these.

| Embedding Name | Definition |
|---|---|
| better | "all-mpnet-base-v2" from SentenceBERT |
| faster | "all-MiniLM-L6-v2" from SentenceBERT |
| glove | "glove-twitter-25" from Gensim |
| fasttext | "fasttext-wiki-news-subswords-300" from Gensim |
| w2v | "word2vec-google-news-300" from Gensim |
| TF-IDF | Simple TF-IDF based embeddings |

Table 7: Embedding used.

## A.3 Models and their Parameters used for the experiments

The parameters mentioned in Table 8 are only those which were taken into consideration for grid-search. Rest all the parameters for that model were left in their default states.

Table 8: Model and their parameters specified for grid-search

| Model | Parameters |
|---|---|
| Logistic Regression | • Penalty: "l1", "l2"

• Regularization Strength("C"): 0.001, 0.01, 0.1, 1.0, 10.0, 100.0, 1000.0

• Solver: "lbfgs", "liblinear"

• Epochs: 100 |
| AdaBoost | Default Parameters Used |
| | Continued on next page |

**Table 8 – continued from previous page**

| Model | Parameters |
|---|---|
| Multi-Layer Perceptron | <ul><li>Activation: "relu", "logistic", "tanh"</li><li>Early Stopping: True</li><li>Initial Learning Rate: 0.0001, 0.001, 0.01</li><li>Epochs: 1000, 5000</li><li>Hidden Layer Size: (150, 150)</li></ul> |
| Perceptron | <ul><li>penalty: "l2", "l1", None</li><li>alpha: 0.0001, 0.001, 0.01, 0.1, 1.0</li><li>eta0: 0.0001, 0.001, 0.01, 0.1, 1.0</li><li>Epochs: 100, 1000, 10000</li></ul> |
| Gaussian Naive Bayes | Var Smoothing: np.logspace(0,-9, num=100) |
| Random Forests | <ul><li>Number of Estimators: 100, 125, 150</li><li>Max Depth: 5, 10, 15, 20</li><li>Min Samples Split: 2, 5, 10</li><li>Bootstrap: True</li></ul> |
| SVM | <ul><li>Regularization Parameter: 1, 0.5</li><li>Kernel: "linear", "poly", "rbf", "sigmoid"</li></ul> |
| XGBoost | <ul><li>Base Estimator Max Depth: 1, 2</li><li>Number of Estimators: 100, 200, 300</li></ul> |
| RNN | <ul><li>Optimizer: RMSProp</li><li>Early Stopping: Monitored over Validation Loss</li><li>Epochs: 100</li></ul> |
| LSTM | <ul><li>Optimizer: RMSProp</li><li>Early Stopping: Monitored over Validation Loss</li><li>Epochs: 100</li></ul> |

**Table 8 – continued from previous page**

| Model | Parameters |
|---|---|
| BERT | <ul><li>"bert-base-uncased" from HuggingFace</li><li>Default HF Parameters</li></ul> |
| HateBERT | <ul><li>"GroNLP/hateBERT" from HuggingFace</li><li>Default HF Parameters</li></ul> |
| BERTweet | <ul><li>"vinai/bertweet-base" from HuggingFace</li><li>Default HF Parameters</li></ul> |

