# OpenReview forum: "Beyond Negativity: Re-Analysis and Follow-Up Experiments on Hope Speech Detection"
_ICLR.cc/2023/TinyPapers — Submitted to Tiny Papers @ ICLR 2023_

### Official Review · Reviewer_eHcs · 2023-03-31

**Confidence:** 2

**Summary Of Contributions:**

The authors tackle the problem of detecting "hopeful" content. They  apply various methods to existing datasets. Some conclusions are drawn.

**Rating:**

Great Start (GS): a submission which meets some of the reviewing criteria but has room for improvement

**Strengths And Weaknesses:**

- Strengths: The problem is relevant, the results potentially interesting

- Weaknesses: The paper is poorly written and exremely obscure.


**Suggested Changes:**

- I do not understand which of the various reported results the authors claim to be their own, as opposed to existing results.

- What is the "Shared Task"?

- What  does "Shared Task Rank 1" mean?

- Why only train BERT-like models  for the second task? Why  not apply  the same models as in the first task? And vice-versa?

- Despite the "tl/dr", I couldn't find any analysis of submissions to the previous Hope Speech Shared tasks. Where is it?

---

> ### Author Response · Authors · 2023-04-20
> **Response to Comments and Suggestions**
>
> We thank the reviewer for the insightful comments and would like to apologize for the lack of clarity. We’ve tried our best to answer all the questions below -
>
> Q - I do not understand which of the various reported results the authors claim to be their own, as opposed to existing results.
>
> The model named “Shared Task Rank 1” is the previous benchmark reported for this dataset, and the rest of the models in Tables 1, 2 are the ones we report for our experiments. We have also reported an exhaustive list of all the models (with their hyperparameters) we tried for the task.
>
> Q - What is the "Shared Task"?
>
> The Shared Task in context to the paper “HopeEDI: A multilingual hope speech detection dataset for equality, diversity, and inclusion” refers to the collaborative challenge organized as a part of the Third Workshop on Computational Modeling of People’s Opinions, Personality, and Emotions in Social Media. A shared task typically involves participants using their expertise and machine learning techniques to address a specific problem or task related to the topic of the workshop, in this case, detecting hope speech related to equality, diversity, and inclusion in social media data. Participants are provided with a dataset, such as the multilingual hope speech detection dataset (HopeEDI) mentioned in the paper, which serves as the basis for the challenge. Participants then compete against each other to develop models or systems that achieve the best performance on the given task, using the provided dataset for training and evaluation.
>
> Q - What does "Shared Task Rank 1" mean?
>
> The model referred to as "Shared Task Rank 1" in Tables 1, 2 represents the benchmark performance previously reported for the dataset.
>
> Q - Why only train BERT-like models for the second task? Why not apply the same models as in the first task? And vice-versa?
>
> The primary factor behind this limitation was the constrained computational power available. We had already tested all models on the initial task, leaving only BERT as the remaining option to try. As a result, we were only able to evaluate BERT on the second task, in conjunction with the top five performing models from Task 1. Additionally, we did not run BERT on Task 1 since Task 2 represents the updated version, and we directed our focused efforts towards it.
>
> Q - Despite the "tl/dr", I couldn't find any analysis of submissions to the previous Hope Speech Shared tasks. Where is it?
>
> Due to space constraints we were unable to perform exhaustive analysis of the submissions but we mention some of them in the second paragraph of the Related Works Section. The exact lines where we analyze them are -
> “In a subsequent paper Chakravarthi & Muralidaran (2021), the models used and their efficacy were detailed, with the most successful models utilizing complex deep learning-based architectures such as BERT Devlin et al. (2019) and XLM-RoBERTa Conneau et al. (2019) in tandem with features like an Inception module Szegedy et al. (2015) over two different sources of embeddings Huang & Bai (2021). Hossain et al. (2021) found traditional ML to lag behind and hence opted for fine-tuning however, our results show that the right embeddings can remove the need for finetuning.”
>
> In hindsight we realize that this paragraph could’ve been written in a better way and therefore we have rewritten it in our latest submission. We hope this makes it clearer.

---

### Official Review · Reviewer_LZkN · 2023-04-04

**Confidence:** 4

**Summary Of Contributions:**

The paper studies computationally efficient methods for hope speech detection in online content, specifically YouTube comments (limited by the choice of the dataset). They clearly demonstrate that traditional machine learning models, combined with appropriate input representations, can perform comparably or even better than complex deep learning models.

**Rating:**

High Potential (HP): a submission which meets the reviewing criteria and has potential to make an impact on the field

**Strengths And Weaknesses:**

Strengths:

1. The paper tackles an important problem in an understudied task, i.e, computationally efficient approaches in hope speech detection without compromising on the quality of the results.

2. The experiments are well designed to provide clear evidence that the studied traditional machine learning models with appropriate input embeddings are comparable to the common deep learning based approaches in the literature.

3. The insights from the experiments could be of significant value in practice, especially due to the fact that simpler models are often easy to deploy at scale of online content and social media networks.

4. Although not available at the time of review, the authors promise to release the code (hopefully by the conference dates) which, along with the detailed experiment specifications in the appendix, would make it easy to reproduce the results.

5. The paper is well-written overall and fits perfectly into the Tiny Paper track! Great job!

Weaknesses:

1. The paper could benefit from a more detailed explanation of the rationale behind the choice of embeddings, the used traditional ML models, and the data augmentation techniques.

**Suggested Changes:**

Minor suggestions:

1. In abstract: I understand the intention of the line "Hope speech refers to YouTube comments/posts ..". However, I think its okay to give general definition of hope speech here and then later make it clear that the experiments are limited to hope speech in YouTube comments/posts since that's what the available datasets contain.

2. There is an unnecessary period after quotation marks in the abstract after insight." and codes should be code.

---

> ### Author Response · Authors · 2023-04-20
> **Response to Comments and Suggestions**
>
> We express our appreciation to the reviewer for their positive review and insightful comments. We acknowledge that further elaboration on the choice of embeddings would have been beneficial, and we have addressed this concern by adding the rationale behind our choices in the appendix.
>
> In response to the suggestions provided by the reviewer, we have revised the definition of hope speech in the abstract and made it broader. In subsequent sections, we have also presented the definition from previous literature. The updated definition we now use is: "Hope speech refers to comments, posts, and other social media messages that offer support, reassurance, suggestions, inspiration, and insight."
>
> We have tried to answer all your queries below -
>
> Q - Why these embeddings?
>
> We chose these 6 different embeddings (present in Table 7) as they cover the most popularly used non-contextual and contextual embeddings. We anticipated SentenceBERT Embeddings to be significantly better than the non-contextual embeddings and we observe the same in our results. We agree that there are several other embeddings which we could’ve tried but the results with these embeddings alone were quite close to the SOTA models and hence we chose to stick with these.
>
> Q - Why these ML Models?
>
> We extensively experimented with various machine learning (ML) models, including XGBoost, RandomForest, and others, which are known for their high performance. Our motivation for utilizing ML models was driven by our belief that deep learning (DL) models often contribute significantly to carbon footprints and should be employed judiciously, only when necessary. We observed that in the "Shared Task" and in general, many practitioners tend to apply DL models to data without proper inspection. However, we found that the same dataset could be effectively addressed using ML techniques with thorough analysis.
>
> Q - Why this Data Augmentation Technique?
>
> Ans - Textual data is not very flexible as compared to data used in vision tasks, which is why the data augmentation techniques for such data aren’t very, so to say, rudimentary. For visual data we can apply techniques like adding gaussian noise, rotating the image, cropping the image et cetera for augmenting and increasing the size of the training set but this is not possible for textual data because it might lead to loss of information and coherence. Hence techniques like Random Word Insertion, Swapping, and Deletion are used which would also account for the loss of coherency.

---

### Meta-Review · Area_Chair_wGMf · 2023-04-07

**Recommendation:** Invite to archive
**Confidence:** 3

**Metareview:**

The two reviewers differ in their evaluations of the paper. I tend to agree more with reviewer `eHcs` after reading the paper. I found the paper lacking in clear descriptions of what was done and what their results were.

Two tables in the main text seem to summarize the results but it is unclear what these tables are displaying. Assuming that "Shared Task Rank 1" indicates the author's results, it seems that the model is performing much more poorly than the other methods (Table 2). According to the authors, one should take Table 2 results more seriously because this table summarizes the Macro F1 score which is "more meaningful" because "the class imbalance would not lead to elevated Weighted F1 scores."


**Summary:**

The paper uses simple classification algorithms to categorize YouTube comments as "hopeful" or not. The comments are converted to word embeddings using various techniques. The problem that's tackled is important. However, the paper is not clear.

**Reason For Not Giving A Higher Recommendation:**

The paper is not CCR.

**Reason For Not Giving A Lower Recommendation:**

The problem is an important and relevant one and improvements in the solutions for detecting "hope speech" can have meaningfully positive effects on society.

---

> ### Author Response · Authors · 2023-04-20
> **Response to Comments and Suggestions**
>
> We express our gratitude to the Area Chair for their valuable suggestions. We acknowledge that our paper requires improvement, and we have made efforts to address the questions raised and weaknesses identified by extensively revising sections and providing additional information in the appendix. We would like to clarify Shared Task Rank 1 is the result of the model which achieved Rank 1 in the Shared Task and not our own model. Our models are the rest of the models mentioned in the table.
>
> Our aim is to make the paper more concise and suitable for consideration in CCR. In terms of reproducibility, we are committed to making all our code publicly available, ensuring that our experiments can be replicated by the research community.
>
> We have also responded to the queries and weaknesses pointed out by both the reviewer and we believe that all of the issues were caused by poor writing instead of technical issues with the work. We hope this revised paper answers all the queries alongside our comments.

---

### Decision · Program_Chairs · 2023-04-10

Invite to archive